# Effect of Sowing Date and Environment on Phenology, Growth and Yield of Lentil (*Lens culinaris* Medikus.) Genotypes

**DOI:** 10.3390/plants12030474

**Published:** 2023-01-19

**Authors:** Lancelot Maphosa, Aaron Preston, Mark F. Richards

**Affiliations:** NSW Department of Primary Industries, 322 Pine Gully Road, Wagga Wagga, NSW 2650, Australia

**Keywords:** biomass, lentil, phenology, sowing date, grain yield

## Abstract

Lentil, an important pulse crop in Australia, is sown soon after the onset of autumn rains and grows mainly under rainfed conditions. This study examined lentil phenological development, growth and grain yield under different sowing dates and environments in New South Wales (NSW). Eight lentil varieties were phenotyped over two years and four sowing times in southern NSW (Leeton, Wagga Wagga and Yanco (one year)) and central western NSW (Trangie). Time of sowing affected important agronomic traits, with a delay in sowing decreasing time to flowering and podding, biomass accumulation, plant height and position of bottom pod. Sowing earlier or later than optimum decreased grain yield. Yield was mainly determined by the number of pods and seeds per plant, with minimal impact from seed weight. Overall, yields were higher in favorable environments such Leeton experiment which received more water compared to the other sites which received less water. Averaged across sowing dates, the slower maturing PBA Greenfield was lower yielding whilst fast maturing varieties such as PBA Bolt and PBA Blitz yielded higher. PBA Jumbo2 is less sensitive to environmental interaction and thus broadly adapted to the diverse environments. Optimum sowing time was identified as the end of April to mid-May.

## 1. Introduction

Lentil (*Lens culinaris* Medikus.) is a diploid (2n = 2x = 14), self-pollinated cool season food legume commonly used as human and animal feed. Two types are produced, the large seeded macrosperma and the small seeded microsperma. The crop has a low and bushy sub-erect or erect architecture and an indeterminate growth habit. This indeterminacy results in prolonged and/or overlapping flowering and podding phases as flowers and pods develop simultaneously on primary, secondary, and tertiary branches. This indeterminacy can offer a potential recovery mechanism after exposure to abiotic stresses if the subsequent environmental conditions are favorable. This has been observed in other pulse crops as well where indeterminacy leads to some recovery from frost damage when adequate soil water is present [1]. Lentil is a valuable rotation crop, especially with cereals as it improves soil health through biological nitrogen (N) fixation [2]. The N fixed by lentils ranges between 0 and 192 kg total N/ha, with an average of 80 kg N/ha and can benefit subsequent cereal crops in terms of yield and protein content [3]. The rotation also allows disease and weed suppression, provides organic matter and moisture retention, thus contributing to favorable cereal crop yields.

Lentil is grown in diverse environments and conditions, such as the subtropical savannah, Mediterranean, and temperate regions. Globally, production has been increasing, partly because of expansion to new production areas such as Australia and Canada, with an average worldwide yield of approximately 1153 t ha^–1^ [4]. Though grown in over 43 countries, Canada accounts for 50% of the world trade [4]. In Australia, lentil is cultivated in an area of over 412,381 ha, producing 525,848 t grain per annum [4]. Despite being a high value crop, lentil is a small crop in terms of area cultivated and production in Australia compared with major crops such as wheat [5,6]. It is mainly produced in medium winter-dominant rainfall zones of Victoria and the mid-north of South Australia on mostly alkaline well-drained soils. Outside these traditionally more favorable cropping areas, lentil production remains small partly due to a lack of adapted genotypes and absence of knowledge on tailored, locally relevant agronomic and management practices. Over the years, the Australian lentil breeding program has been selecting for early maturing varieties to minimize the impact of abiotic stresses and match developmental stages with the available resources [7,8].

In Mediterranean environments such as Australia, lentils are generally sown in late autumn or early winter and emerge into cool temperatures and short days but as the season progresses towards spring, temperatures rise and daytime lengthens. Therefore, it tends to experience low temperatures including severe frost events during the vegetative stage and high temperature and drought stress during the reproductive phase. The growing season ranges from mid-April to mid-December in these environments. The optimum growth temperature range for lentil is 10–30 °C [9,10], and maximum temperatures above 35 °C are highly detrimental for its growth and yield [11,12]. A decrease of 0.11% and 0.13% in grain set and yield, respectively, has been observed for every degree above 32 °C [13]. Low temperatures or frost slow phenological development, resulting in a longer vegetative period, pod abortion and decrease in yield. Heat stress affects phenological development and can reduce the growth rate, shorten the flowering, podding and overall growth duration, and reduce biomass accumulation, and during the reproductive phase, it affects pollen, ovule and stigma properties, and thus compromising grain weight, composition and yield [11,12,14,15,16,17]. Moisture stress results in reduced grain number and weight, with the reproductive stage more sensitive than the vegetative one [18,19]. The impact of heat stress or moisture deficient individually on yield is less than when they occur simultaneously [13,19].

Lentil production could potentially expand to new environments of southern and central western regions of NSW where other pulse crops are grown. However, this expansion is hindered by limited identification of adapted genotypes and optimal agronomic management information such as sowing times. Staggered sowing in multiple locations across years is an established approach to assess crop adaptation in new production areas. Testing the matrix of genotype × management practices such as sowing depth, plant density and sowing date allows identification of optimal combinations [7,20]. The objectives of this study were to a) examine the influence of sowing time on the timing, duration of key lentil phenological growth phases, growth and grain yield and b) identify varietal performances across contrasting environments of southern and central western NSW.

## 2. Results

### 2.1. Temperature at Sowing and Day Length

Mean daily temperature at sowing, important for lentil germination, varied between the sites (Table 1), progressively decreasing as sowing date (SD) was delayed and had a wider range in 2019 than 2018. Temperatures were higher at the central western site (Trangie) than the southern sites for later sowing dates (SD3 and SD4).

Days were slightly longer between April and August at TARC and shorter in the other months (Figure 1) compared with the southern sites but these differences were minimal. Day length did not differ between 2018 and 2019 at the same location and was largely similar at the southern sites.

### 2.2. Effect of Sowing Date on Phenology

Sowing date affected the timing and duration of key lentil growth stages (Table 2) but the effect of sowing date on establishment was inconsistent across years and sites. There was no effect on establishment in WWAI2018, WWAI2019 and YAI2018. In TARC2018, establishment was lowest in SD3, while in LFS2018 it was lowest in SD3 and SD4. In TARC2019, establishment was lowest in SD1 and in LFS2019 it was lowest in SD1 and SD2. In TARC2018, establishment was poor overall compared to the target value of 120 plants/m^2^ while at YAI2018, LFS2018 and WWAI2019 establishment was above the target value.

Days to emergence were shorter in SD1 at all sites and progressively became longer as sowing was delayed, ranging from four days at WWAI2018 to 19 days at LFS2019. At LFS and WWAI in both years, days to flowering and podding were longer in earlier sowing dates (SD1 and SD2). At TARC2018 and YAI2018, flowering was longer for SD3 and in TARC2019, it was longer from SD2 to SD4. In TARC2018 time to podding was longer in SD3, while at YAI2018 it was longer in SD2 and in TARC2019 it was longer from SD2 to SD4. At all sites, days to flowering tended to decrease as the mean daily temperature increased. All genotypes flowered earlier at TARC, the warmest location with longer day length in the colder winter months, where it took less than 100 days in both years. YAI2018, which was dry compared with the adjacent LFS2018, also had a shorter vegetative phase. The overall phenological development (duration of vegetative phase, flowering, and podding) decreased with delayed sowing time. The exceptions were flowering and podding durations at TARC and WWAI in 2019. PBA Blitz was the earliest to flower at both sites and Nipper and PBA Greenfield were the slowest (Appendix A).

### 2.3. Effect of Sowing Date on Architecture, Biomass and Related Components

Time of sowing largely affected crop architecture and biomass accumulation (Table 3). Branch number ranged from 4 (TARC2018 and LFS2019, SD4) to 12 at LFS2018 (SD1) but was not affected by time of sowing at WWAI in both years. At the other experiments, there were more branches at earlier sowing (SD1 and/or SD2) than when sowing was delayed. Bottom pod height ranged from 11.15 at TARC2019 (SD1) to 25.60 cm (LFS2019 SD1) but was not influenced by time of sowing in YAI2018 and LFS2018. However, in all the other sites it was higher at earlier sowing (SD1 and/or SD2) than when sowing was delayed. Top pod height ranged from 21.66 at TARC2019 (SD4) to 50.99 cm at LFS2019 (SD1) and was not influenced by time of sowing only at YAI2018. In all the other sites, it was higher at earlier sowing (SD1 and/or SD2) than when sowing was delayed. Total biomass accumulated at harvest was not influenced by time of sowing only at YAI2018. The lowest biomass was accumulated at the latest sowing time (SD4) in all the other experiments. It ranged from 1.883 at TARC2019 SD4 to 7.562 t/ha at LFS2018 SD1. Harvest index was not affected by time of sowing only at TARC2019. In the other sites, it increased when sowing was delayed except in TARC2018 where it decreased. It ranged from 0.06 at WWAI2019 SD1 to 0.50 at WWAI2018 SD4.

### 2.4. Effect of Sowing Date on Grain Yield and Yield Components

Time of sowing largely affected total grain yield and yield components (Table 4). The number of filled pods at different sowing times was different at TARC2018, TARC2019 and LFS2019, and not in the other four experiments. There were more filled pods in early sowing (SD1) than late sowing (SD4) and ranged from a high of 97 in TARC2019 SD1to a low of 12 in WWAI2019 SD1. There was no difference in the number of unfilled pods at WWAI2019 and LFS2019. The number of unfilled pods was as low as two in WWAI2019 (SD1, SD2 and SD4) with a high of 24 in TARC2019 SD1. The total number of pods across sowing times was different only in the TARC experiments. Across experiments, it ranged from 14 at WWAI2019 (SD1) to 115 at TARC2018 (SD1). Seeds per pod were different in all the experiments except LFS2019 but were on average close to one.

Seeds per plant were different at TARC2018, TARC2019 and YAI2018. They ranged from 12 at YAI2018 and WWAI2019 SD1 to 114 at TARC2018 SD1. There were fewer seeds at SD1 at YAI2018 while at the TARC experiments there were more seeds at earlier sowing. Seed weight (100 seed weight), which ranged between 3 and 5 g was different at TARC2018, WWAI2019, LFS2018 and YAI2018. At TARC2018 and LFS2018, higher seed weight was obtained at earlier sowing while at WWAI2019 early sowing resulted in low seed weight. At YAI2018, highest seed weight was obtained at SD2 and SD3.

Final grain yield was different in all the experiments and was very low at WWAI2019 SD1 (0.187 t/ha). The highest yield of 2.433 t/ha was obtained at LFS2018 SD3. In TARC2018, early sowing (SD1 and SD2) resulted in higher yield compared to late sowing, while in TARC2019 the first three sowing times yielded similarly. In WWAI2018, SD2 and SD3 yielded highest, while in WWAI2019 SD3 yielded highest. In LFS2018 and YAI2018, SD3 and SD4 yielded highest, while in LFS2019 SD2-SD4 yielded highest. Correspondingly, machine yield was lowest at WWAI2019 SD1 at 0.220 t/ha and highest at LFS2018 SD3 at 2.158 t/ha.

### 2.5. Genotypic Yield Responses to Sowing Time

There were different varietal responses and interactions with time of sowing. At Wagga Wagga, PBA Greenfield was the lowest yielding variety when averaged across SDs in both years. In 2018, mean grain yield averaged across SDs ranged from 1.280 t/ha for PBA Greenfield to 1.550 t/ha for PBA Ace and in 2019 it ranged from 0.480 t/ha for PBA Greenfield to 0.700 t/ha for Nipper but showed G × SD interactions. In 2019, all varieties except the highest yielding Nipper (0.700 t/ha) yielded similarly at Wagga Wagga. Nipper demonstrated broad adaptation at Wagga Wagga with the highest yield when averaged across all sowing dates and was the best performing variety when sown early (SD1). At Leeton, PBA Greenfield was lowest yielding variety when averaged across SDs in both years. Mean grain yield averaged across SDs ranged from 1.280 t/ha for PBA Greenfield to 2.520 t/ha for PBA Bolt and showed G × SD interactions in 2018, and from 0.610 t/ha for PBA Greenfield to 1.330 t/ha for PBA Bolt and showed G × SD interactions in 2019.

At Yanco (YAI2018), mean grain yield averaged across SDs it ranged from 0.760 t/ha for PBA Greenfield to 1.320 t/ha for PBA Bolt but showed no G × SD interactions. At Trangie, in 2018 mean grain yield averaged across SDs it ranged from 0.740 t/ha for PBA Greenfield to 1.390 t/ha for PBA HallmarkXT and did not show G × SD interactions. In 2019, at Trangie, when averaged across SDs yield ranged from 0.570 t/ha for PBA Blitz to 0.920 t/ha for PBA Hurricane XT and showed G × SD interactions.

Overall yields were higher in the Leeton experiment which was heavily pre-irrigated compared to the other two sites. Frosts were more severe at Wagga Wagga than at Leeton. 

#### 2.5.1. G × E Interactions

The AAMI biplot (Figure 2) is useful for exploring the G × E interactions. Both PCA1 (*p* < 0.001) and PCA2 (*p* = 0.0398) are significant, with PC1 axis explaining 60.79% and PC2 axis 20.78% of the G × E interaction sum of squares (Figure 2). Therefore, the principal component axis explained 81.57% of the G × E interaction. The genotypes did not cluster together in the AMMI biplot and thus did not behave similarly across environments. Genotype PBA Jumbo2 is closer to the origin (less sensitive to environmental interaction and thus broadly adapted) than PBA Blitz, PBA Bolt, Nipper and PBA Greenfield which are specifically adapted. Environments WWAI2018 and TARC2019 clustered together and thus influence genotypes in a similar way. There was a positive correlation between YAI2018, with LFS2018 and WWAI2019; LFS2019 with TARC2018 and LFS2018. There was no correlation between YAI2018 and LFS2019; a negative correlation between YAI2018, and TARC2018; TARC2018 and WWAI2019. Environments LFS2018, LFS2019 and TARC2018 are further away from the origin.

#### 2.5.2. Classification of Environments

The GGE biplot (Figure 3) shows the performance of different genotypes in different environments. The GGE biplot characterizes the seven experiments into three mega-environments, one comprising WWAI2019 (ME1), the other LFS2019 and TARC2018 (ME2), and the other comprising the environments WWAI2018, TARC2019, LFS2018 and YAI2018 (ME3). Genotype Nipper yielded higher in the sector comprising ME1, while PBA Jumbo2, PBA HurricaneXT and PBA HallmarkXT yielded higher in the sector comprising ME2 and genotypes PBA Bolt yielded high in the sector comprising ME3. PBA Blitz, PBA Greenfield and PBA Ace showed environmental sensitivity and were in sectors not comprising any of the mega-environments.

## 3. Discussion

Environmental conditions such as temperature, moisture availability and daylength varied among the different environments/locations/sites and at different phenological stages and were the main drivers of lentil development. Day length and temperatures increase while moisture diminishes as the season progress in these winter-sown environments. However, for lentils it is generally accepted that temperature has a much bigger impact on development (time to flowering) than photoperiod [21,22]. There are potentially other confounding factors such as relative humidity and radiation which might alter the phenological response [23,24]. Generally, in Australia, days are longer in the north than in the south as was observed in this study but unlike temperature and moisture availability that varied widely between years, day lengths were similar in the two years for each site. Prolonged germination was observed for the late sowing date (SD4) in both years and all sites, as a result of the low soil temperature at sowing, with the only exceptions being LFS2018 and YAI2018 where germination was delayed at SD2 and SD3 (Table 2). This is similar to findings in another study where late sowing delayed lentil emergence by 8 days [25] and to a chickpea study conducted under the same conditions as this study [20]. However, all the experiments in this study were pre-irrigated and/or irrigated immediately after sowing with different amounts of water; therefore, eliminating any potential moisture stress, the differences in emergence are attributed to soil temperature.

The reduction in the number of days to flowering and maturity when sowing is delayed past the optimum or under unfavorable environments is generally known. Thus, the observations in this study are in agreement with another study where a 1.5 °C increase in temperature accelerated flowering by between 2.5 and 18.1 days and a 0.1 hour increase in daylength altered flowering by between 0.6 and 5.4 days in winter-sown lentils under Mediterranean conditions [24]. At TARC, where days are longer and mean temperatures are higher, flowering and overall phenological development was accelerated compared with the southern sites. Selection for early flowering and maturity is desirable in shorter Mediterranean environments to avoid late season stresses but should also consider the risk of frost occurrence. While differences in the vegetative duration were large, they were minimal for the reproductive phase, an indication that adaptation is largely driven by the vegetative phase. Flowering time determines vegetative phase duration, and the onset of the reproductive phase and ultimately the climatic conditions the crop will be exposed to for the remaining duration of crop growth. Later in the growing season, photoperiod becomes non-limiting, and consequently temperature and soil water become the major climatic variables responsible for the rate of progress from flowering to physiological maturity. The findings of the present study are consistent with previous ones where the impact of temperature on lentil development and length of growth phases has been demonstrated [23,25].

Though lentil is fairly drought resistant, the dry conditions due to limited irrigation at YAI resulted in accelerated plant growth compared with LFS with the same temperature, rainfall, day length and soil characteristics. Soil moisture availability greatly delays pulse crop maturity [19,20,25]. Moisture limitation at YAI would have lowered growth rate and leaf expansion, caused early leaf senescence thus accelerating time to physiological maturity and shortening the podding duration [12,17,26,27,28]. Early flowering and maturing varieties are capable of escaping late season heat and drought stress and this could enable lentil production to expand to marginal, drier and short-season areas [18,29,30,31,32]. These early maturing varieties such as PBA Blitz and PBA Bolt yield better than later varieties especially in dry years, short season low rainfall environments or years where high temperatures cause premature drying of the crop [30] and were also high yielding in this study. Furthermore, due to early maturity and shorter growing season duration they might leave more water in the soil at the end of the cropping season for subsequent crops [33]. However, in favorable seasons, they might not reach their full yield potential due to substantial water being left in the soil profile. Additionally, a relationship between growth rate and sensitivity to other abiotic stresses has been reported with slow growing lentils more sensitive to salt than those growing rapidly [34].

Environmental conditions such as drought, frost or heat stress affect lentil growth and if they occur at sensitive stages further affect biomass and the components of seed yield, with effects more severe if they occur at the same time [26]. Limited moisture availability has been shown to decrease yield, and at times by up to 28% [13,29]. Drier conditions at YAI resulted in less biomass accumulation and grain yield compared to LFS. Drought at the YAI would have potentially reduced yield through decreasing the number of flowers produced, increased seed abortion (empty pods), accelerated the grain-filing rate and remobilization of N from vegetative parts to grain [27,28,35]. However, it has been observed that mild water deficit stimulates flower production in some genotypes [28]. High temperatures lead to a reduction of seed weight due to a reduction in the seed filling rate and/or shortened podding duration [12,17,26]. The decrease in rate and duration of grain filling is likely to result in reduced seed weight through limiting sucrose importation, reduced starch deposition to the developing grain, inhibition of starch synthesizing enzymes and thus impairing starch accumulation in the developing grain [19,26,36]. However, the effects are not universal, for example in other studies, heat stress [15] and drought stress [27] did not have an effect on seed size, a primary determinant of market quality and price. This might largely be due to differences in the timing of the stress in relation to the crop phasic development.

Due to the indeterminant growth habit which creates competition between pods and vegetative parts for photosynthates, the harvest index was lower in SD1, and this is due to a disproportionately large vegetative biomass [37]. However, the indeterminant growth habit might favor production of more flowers and the maintenance of pod and seed set when plants subsequently experience favorable conditions following exposure to stressful ones. As was also observed in SD3 in this study, in addition to reducing lodging, moderate biomass and reduced branching, results in improved harvest index and yield in lentil [37,38]. This might be because moderate biomass results in more light penetration, supports more pods and is optimal for balanced source-sink dynamics. The low biomass at SD4 might be due to the shorter growth period not allowing sufficient time to accumulate resources as plant growth was accelerated. Early sowing (SD1) and the accompanying longer vegetative phase resulted in higher biomass production, a finding observed in lentil [29] and in beans, another pulse crop [39]. However, the higher biomass at earlier stages negatively impacts yield due to longer exposure to early season frost events. There is a need to develop management systems that increase HI through allowing efficient conversion of biomass into grain yield. In addition, the lower yield at SD1 is likely because of excessive biomass production, potentially resulting in higher plant transpiration and the large biomass consumes resources such as water but also shades the lower flowers and prevent light penetration, resulting in increased flower abortion and/or limited development. The higher biomass at SD1 also increases the incidences of lodging as was observed in this study at the sites in which lodging score was taken (results not shown). Warmer and drier conditions due to late sowing may reduce the capacity of plants to fill seeds [39]. The decrease in yield under high temperatures is partly because of reduced number of flowers, pod set, pod and flower abortion and fewer filled pods [14,27].

Accumulated biomass can be related to plant architecture to some extent, with taller plants accumulating more biomass and/or weighing more. Of particular importance to breeding programs and growers is bottom pod height as it is important for machine harvestability, speed of harvest and overall harvest losses. PBA Bolt has previously been shown to have good pod retention and bottom pod height, which are key for harvesting crops exposed to abiotic stresses [30], a finding confirmed in this study. Late sowing and associated late season stresses reduced branch number, both bottom and top pod heights, a finding observed in another study [40]. Variation in branch and pod number has also been previously reported in lentil [37,41]. The number of pods, seeds per plant and harvest index have been shown to be the main determinants of grain yield in lentils [41]. Sowing date had a significant effect on lentil grain yield in our study, and the yield response was through the various yield components. In this study, like in most other studies, the number of pods per plant, grain number, branch number, and grain weight were all shown to be determinants of yield [37,40,42]. Previous studies have shown a reduction in seed weight and seed number due to increased pod losses and fewer filled pods [12,19,26]. Impact of sowing time on yield and yield components has also been observed in other pulses such as chickpea [43] and soybean, [44]. The yield reduction in response to mismatched sowing time can be either through reduction of pod number, seed number or weight but it depends on the timing, severity and duration of the weather conditions at critical growth phases. If plants reduce grain number in response to stress, source-sink dynamics are altered and the existing assimilate gets channeled towards increasing the grain size of the maintained grains. This compensatory effect is responsible for the often-observed negative correlation between these traits in most crops. In lentil, the small seed sized genotypes were observed to produce more flowers, pods and seeds compared to large seed sized ones [27,28].

The environmental correlations and mega-environment classification did not reflect geographic locations indicating that factors such as altitude and photoperiod probably have limited effect on lentil grain yield. This would mean other conditions such as temperature and moisture availability played a bigger role in the classification and thus were the main drivers of lentil grain yield. It is generally accepted that temperature plays a larger role than photoperiod in lentil adaptation [21,22]. The genetic diversity of genotypes was confirmed as they behaved differently across environments and did not cluster together in the AMMI biplot. Furthermore, genotype PBA Jumbo2 was shown to be broadly adapted to environments of central western and southern NSW unlike PBA Blitz, PBA Bolt, Nipper and PBA Greenfield which are specifically adapted.

The multi-year and multi-location approach conducted in this study provides better prediction of genotype performance across diverse environments including correlations between environments with similar growing conditions. Different sowing dates outside the recommended sowing window can be used to simulate and/or test sensitivity and adaptability and have been used to examine lentil performance, and to devise better overall management practices, such as matching variety to production environment [2,7,12,14,25,33]. Sowing in the desirable window results in the mismatching of sensitive stages with the occurrence of abiotic stresses allowing phenological development to occur under favorable conditions. Variation in adaptation was observed in this study thus providing opportunities to recommend suitable varieties and sowing times for the different locations. Significant genotypic differences in phenological development in this study confirm that high genetic diversity exists among varieties used and provides opportunities to recommend suitable varieties and/or sowing times for the different regions. 

No information exists as to the optimum sowing time for lentil in southern and western NSW, and thus the present study was undertaken to better fit cultivars to production areas. Results presented here indicate that sowing around the mid-May period in southern NSW and the mid-late April period in central NSW gives the varieties tested the best opportunity to avoid abiotic stresses and allows efficient conversion of biomass to grain yield. Early sowing or longer maturing varieties such as PBA Greenfield risk greater exposure to potential frost damage and late season adverse conditions such as terminal drought and heat stress. However, it is important to consider that the 2018 and 2019 seasons in southern and central western NSW were very challenging for growing crops as they experienced low autumn and winter rainfall, and low temperatures early in the season. The below average seasonal rainfall across the two years identified genotype responses in dry seasons but not under ordinarily normal growing seasons. 

## 4. Materials and Methods

### 4.1. Experiment Locations, Climatic Data and Management

Experiments were conducted at Trangie (31.99° S, 147.95° E) at the Trangie Agricultural Research Centre (TARC), in central western NSW, Wagga Wagga (35.05° S, 147.35° E) at the Wagga Wagga Agricultural Institute (WWAI), Leeton (34.59° S, 146.36° E) at the Leeton Field Station (LFS) and Yanco (34.61° S, 146.41° E) at the Yanco Agricultural Institute (YAI) in southern NSW in 2018 and 2019 (Yanco 2018 only). Experiments were started at similar times at all sites, with fortnightly sowing from mid-April to end of May. Experiment site details, seasonal rainfall, supplementary irrigation and overall management practices have been described previously in a companion chickpea paper under the same conditions [20]. The long-term climatic conditions were obtained from the Australian Bureau of Meteorology (BOM) website (http://www.bom.gov.au (accessed on 5 June 2020)) and have been previously published [20] and are thus provided here as Appendix A. The long-term average growing season rainfall at Trangie is 248 mm; however, only 137 mm and 45 mm was received in 2018 and 2019, respectively. The long-term average growing season rainfall at Wagga Wagga is 322 mm; however, only 153 mm and 193 mm was received in 2018 and 2019, respectively. Likewise, the 2018 growing season rainfall at Leeton/Yanco of 87 mm was well below the 193 mm long-term average but improved in 2019 to 160 mm. At all sites and in both years, supplementary water was applied pre-sowing and during the season to assist with establishment and overall crop growth. Day length (hours of plant photosensitive light) was calculated using APSIM model by incorporating, day of year, latitude and civil twilight with a sun angle of 6 degrees below the horizon.

### 4.2. Plant Material

Eight diverse lentil genotypes consisting of released varieties (Table 5), were used to evaluate phenological development, across four sowing dates. The genotypes were selected based on their maturity type (early, mid and late), seed classification (color and size), and herbicide tolerance (imidazolinone herbicides).

### 4.3. Experiment Design

The experiment was a split-block design with three replicates, with sowing date as main plot and genotypes randomized within plots. Seeds were sown 3–5 cm deep to ensure good emergence and to avoid damage from pre-emergent herbicides. The seeding rate was adjusted to achieve a target sowing density of 120 seeds/m^2^ which is the optimal density for a range of Australian environments [7]. In both years, a five (TARC) or six-row cone seeder was used for sowing. The row spacing was 0.33 m at TARC, 0.3 m at WWAI and 0.25 m at both LFS and YAI, resulting in a plot area of 16.5 m^2^ at TARC, 21.6 m^2^ at WWAI, 15 m^2^ at LFS and YAI. At sowing, a *Rhizobium* group N peat-based inoculant (New Edge Microbials, Albury, Australia) was made into a water slurry and injected into the furrow at a rate of 80 L/ha. Local best management practices were followed including hand weeding and applying registered herbicides, fungicides and insecticides to minimize the effects of weeds, diseases and insect pests.

### 4.4. Phenological Measurements

Phenological measurements were undertaken as described previously for chickpea [20] and adopted from widely published literature. Emergence date (D50%emer) was recorded as the day when 50% of the targeted population had emerged, counting plants in the inside rows in two separate m^2^ quadrats. After emergence a reference area was marked within the plots by counting 20 plants in an inside row. All subsequent phenological measurements were taken from this reference area. Days to 50% flowering (D50%F) were recorded as the date when 50% of the 20 plants (i.e., 10 plants) within the reference area had at least one open flower. End of flowering was recorded when flowers from all plants within the reference area had withered or dropped. Similar measurements were taken for days to pod initiation (D10%P), days to 50% podding (D50%P), and days to physiological maturity (DTPM, defined as the date when 95% of the pods in a plot changed to a yellow brown color). These measurements were then used to calculate vegetative (VD), flowering (FD) and podding (PD) durations.

### 4.5. Measurements at Physiological Maturity

Plant height was measured in situ, from the base to the to the top of the plant, repeated three times along the plot. Ten plants were taken at random from each plot to measure plant components including: top and bottom pod height, number of filled/unfilled (viable/unviable) pods, total pods, pod weight, branch number, seeds per plant and seeds per pod. Two harvest cuts to calculate yield and yield components were taken from a m^2^ area of the inner rows, excluding border rows at least 1 m from the ends of each plot. Total above ground biomass (t/ha), grain yield (t/ha), harvest index, seed number and weight (g) were calculated from this sample. Harvest Index (%) was calculated as (grain yield/total shoot dry weight) × 100. The remainder of the whole plot was mechanical harvested, and mechanical harvest yield adjusted for area which had been cut out.

### 4.6. Statistical Analysis

Statistical analysis was done using the Restricted Maximum Likelihood (REML) spatial linear model algorithm in GenStat 20th Edition [45] for individual experiments (single site and year), to understand the effect of genotype, sowing date and the interaction between them. In the model, the main effects and their interactions were fitted as fixed, while row, column, replication, main and subplot were fitted as random. This allowed for the estimation of spatial trends within the field. The predicted means for grain yield generated from the REML model were used to test the environmental correlations, G × E interactions, genotype adaptability/stability using the additive main effects and multiplicative interaction (AMMI) model analysis. Furthermore, the predicted means for yield were used to characterize the environments using the genotype main effects (G) and genotype by environment interaction (GxE) GGE model.

## Figures and Tables

**Figure 1 plants-12-00474-f001:**
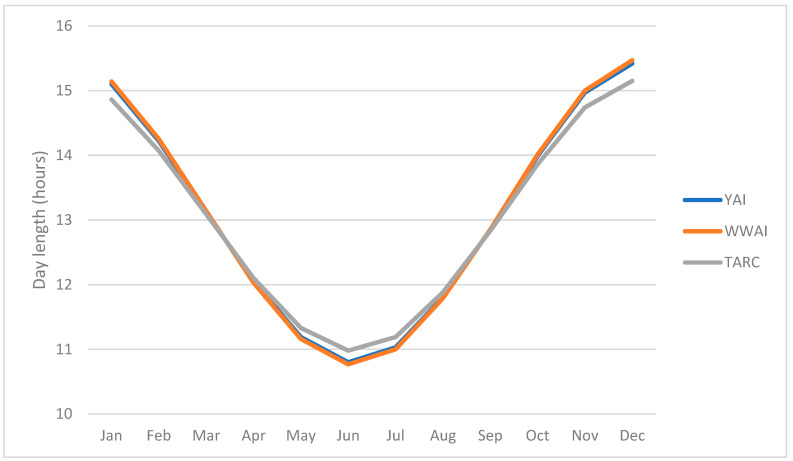
Day length (photoperiod) in hours at YAI (and LFS), WWAI and TARC.

**Figure 2 plants-12-00474-f002:**
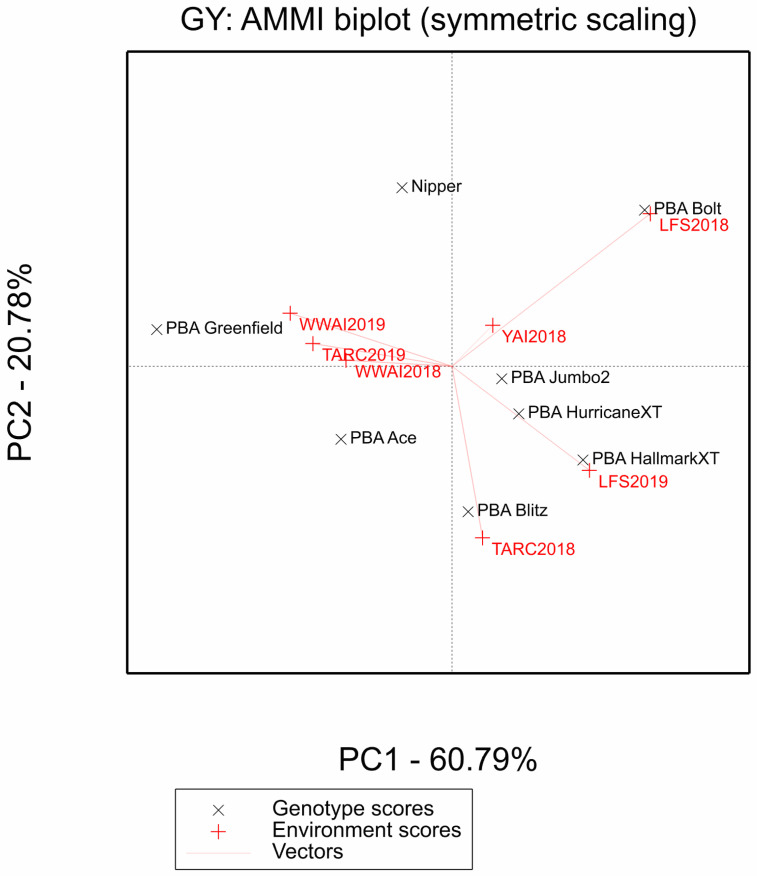
AAMI biplot for grain yield showing the correlation between environments and overall genotype stability and adaptability. At origin GEI = 0. Acute angle = positive correlation; right angle = no correlation and obtuse angle = negative correlation. TARC18 = Trangie Agricultural Research Centre 2018 experiment; TARC19 = Trangie Agricultural Research Centre 2019 experiment; WWAI18 = Wagga Wagga Agricultural Institute 2018 experiment; WWAI19 = Wagga Wagga Agri cultural Institute 2019 experiment; LFS18 = Leeton Field Station 2018 experiment; LFS19 = Leeton Field Station 2019 experiment; and YAI18 = Yanco Agricultural Institute 2018 experiment.

**Figure 3 plants-12-00474-f003:**
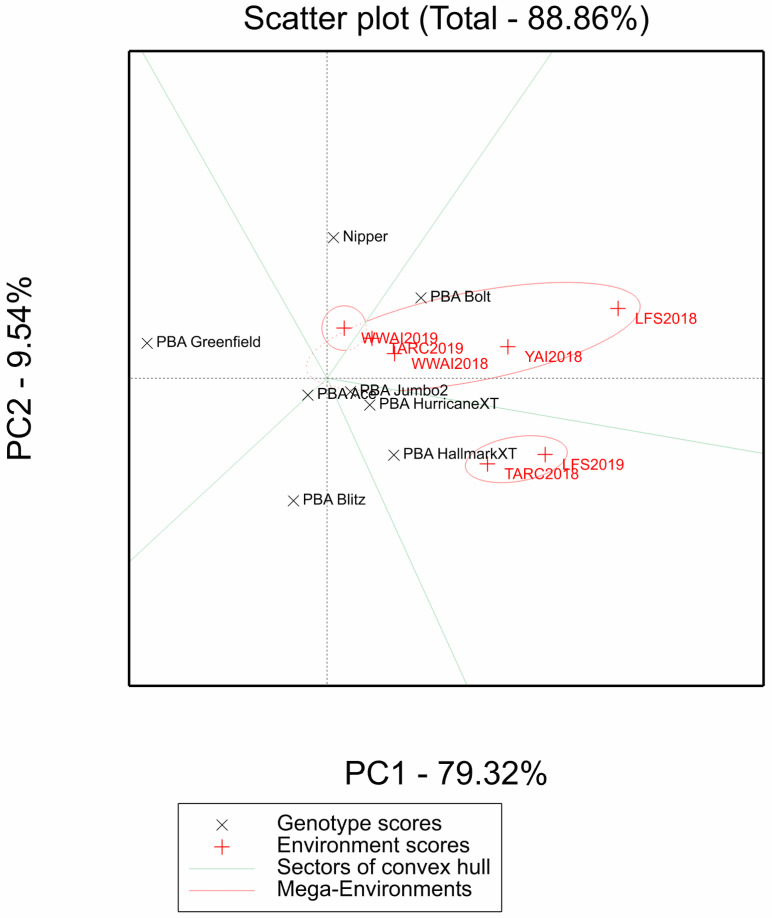
GGE biplot for grain yield showing different vectors and three mega environments and variety performance in the respective vectors and mega environments. The overlapping environments in ME2 are LFS2019 and TARC2018. The overlapping environments in ME3 are WWAI2018 and TARC2019. TARC18 = Trangie Agricultural Research Centre 2018 experiment; TARC19 = Trangie Agricultural Research Centre 2019 experiment; WWAI18 = Wagga Wagga Agricultural Institute 2018 experiment; WWAI19 = Wagga Wagga Agricultural Institute 2019 experiment; LFS18 = Leeton Field Station 2018 experiment; LFS19 = Leeton Field Station 2019 experiment; and YAI18 = Yanco Agricultural Institute 2018 experiment.

**Table 1 plants-12-00474-t001:** Mean daily temperature (°C) for each sowing date (SD) at Trangie Agricultural Research Centre (TARC), Wagga Wagga Agricultural Institute (WWAI), Leeton Field Station (LFS) and Yanco Agricultural Institute (YAI) in 2018 and 2019.

	2018				2019			
	SD1 (Mid-April)	SD2 (Late April)	SD3 (Mid-May)	SD4 (Late May)	SD1 (Mid-April)	SD2 (Late April)	SD3 (Mid-May)	SD4 (Late May)
TARC	19.4	16.1	14.1	17.2	21.7	18.4	17.4	8.8
WWAI	15.5	13.3	11.3	13.5	21.8	15.5	9.8	7.5
LFS/YAI	18.5	15.2	12.9	16.4	22.9	16.0	10.4	7.4

**Table 2 plants-12-00474-t002:** Influence of sowing date on key lentil development phases. SD = Sowing Date, Est = establishment; D50%emer = days to 50% emergence; D50%F = days to 50% flowering; D10%P = days to 10% podding; D50%P = days to 50% podding; DTPM = days to physiological maturity; VD = vegetative duration; FD = flowering duration; PD = podding duration; * = data not collected. TARC2018 = Trangie Agricultural Research Centre 2018 experiment; WWAI2018 = Wagga Wagga Agricultural Institute 2018 experiment; LFS2018 = Leeton Field Station 2018 experiment; YAI2018 = Yanco Agricultural Institute 2018 experiment; TARC2019 = Trangie Agricultural Research Centre 2019 experiment; WWAI2019 = Wagga Wagga Agricultural Institute 2019 experiment; and LFS2019 = Leeton Field Station 2019 experiment.

Experiment	SD	Est (Plants m^2^)	D50%	D50%F	D10%P	D50%P	DTPM	VD (Days)	FD (Days)	PD (Days)
			Emer							
TARC2018	1 (mid-April)	44	*	94	100	111	169	*	49	58
	2 (late April)	41	*	95	98	108	160	*	39	53
	3 (mid-May)	34	*	98	105	111	154	*	28	43
	4 (late May)	46	*	91	101	105	148	*	25	43
	*p* value	0.006	*	<0.001	<0.001	<0.001	<0.001	*	<0.001	<0.001
	l.s.d. (*p* < 0.05)	6.586	*	1.956	3.034	1.545	2.801	*	2.679	2.854
TARC2019	1 (mid-April)	82	6	85	96	101	180	69	75	84
	2 (late April)	118	8	91	97	107	169	73	55	71
	3 (mid-May)	118	10	95	100	110	157	74	48	56
	4 (late May)	116	16	94	97	104	141	69	37	44
	*p* value	<0.001	<0.001	<0.001		<0.001	<0.001	0.046	<0.001	<0.001
	l.s.d. (*p* < 0.05)	9.207	0.868	3.306	ns	3.555	1.79	4.224	6.553	5.917
WWAI2018	1 (mid-April)	126	4	129	148	152	195	125	50	43
	2 (late April)	117	11	124	136	140	180	112	41	40
	3 (mid-May)	123	18	120	129	132	167	102	30	35
	4 (late May)	109	18	112	119	122	156	94	26	34
	*p* value	0.066	<0.001	<0.001	<0.001	<0.001	<0.001	<0.001	<0.001	<0.001
	l.s.d. (*p* < 0.05)	ns	0.224	1.743	0.792	0.584	0.365	1.887	1.553	0.755
WWAI2019	1 (mid-April)	133	7	127	156	160	192	102	61	36
	2 (late April)	138	10	127	144	146	179	115	33	36
	3 (mid-May)	148	12	118	131	133	168	102	33	37
	4 (late May)	139	14	110	120	122	156	91	28	37
	*p* value	0.058	<0.001	0.002	<0.001	<0.001	<0.001	0.01	0.07	0.473
	l.s.d. (*p* < 0.05)	ns	0.293	6.405	1.143	0.841	0.938	10.06	ns	ns
LFS2018	1 (mid-April)	145	7	119		147	195	112	49	48
	2 (late April)	148	14	117		140	181	103	37	41
	3 (mid-May)	135	14	112		132	164	98	30	33
	4 (late May)	135	11	104		120	152	93	27	32
	*p* value	<0.001	<0.001	<0.001		<0.001	<0.001	<0.001	<0.001	0.001
	l.s.d. (*p* < 0.05)	4.707	0.232	1.221		2.471	2.309	1.037	2.323	2.983
LFS2019	1 (mid-April)	107	7	136	151	160	189	104	55	38
	2 (late April)	106	12	129	138	145	173	105	34	35
	3 (mid-May)	119	12	117	126	132	161	99	26	35
	4 (late May)	125	19	109	114	117	149	83	23	35
	*p* value	<0.001	<0.001	<0.001	<0.001	<0.001	<0.001	<0.001	<0.001	0.002
	l.s.d. (*p* < 0.05)	7.873	0.717	3.7	1.027	1.463	1.087	3.607	2.652	1.192
YAI2018	1 (mid-April)	142	7	104	120	129	189	97	64	60
	2 (late April)	136	14	104	120	135	173	91	46	38
	3 (mid-May)	140	14	108	123	129	158	94	31	29
	4 (late May)	132	11	104	115	121	148	93	25	27
	*p* value	0.346	<0.001	0.035	<0.001	<0.001	<0.001	0.003	<0.001	<0.001
	l.s.d. (*p* < 0.05)	ns	0.132	2.719	2.152	3.108	0.73	2.659	4.181	3.502

ns = not significant (l.s.d. not calculated).

**Table 3 plants-12-00474-t003:** Influence of sowing date on lentil architecture and biomass characteristics. * = data not collected. TARC2018 = Trangie Agricultural Research Centre 2018 experiment; WWAI2018 = Wagga Wagga Agricultural Institute 2018 experiment; LFS2018 = Leeton Field Station 2018 experiment; YAI2018 = Yanco Agricultural Institute 2018 experiment; TARC2019 = Trangie Agricultural Research Centre 2019 experiment; WWAI2019 = Wagga Wagga Agricultural Institute 2019 experiment; and LFS2019 = Leeton Field Station 2019 experiment.

Experiment	SD	Branch Number	Bottom Pod Height (cm)	Top Pod Height (cm)	Plant Height (cm)	Dry Matter (t/ha)	Harvest Index
TARC2018	1 (mid-April)	7	15.25	29.46	*	3.628	0.43
	2 (late April)	5	13.63	24.30	*	3.595	0.42
	3 (mid-May)	6	12.83	23.25	*	2.767	0.35
	4 (late May)	4	13.00	21.99	*	1.953	0.34
	*p* value	<0.001	0.016	<0.001		<0.001	<0.001
	l.s.d. (*p* < 0.05)	1.307	1.473	2.283		0.428	0.026
TARC2019	1 (mid-April)	7	12.99	31.75	32.00	2.758	0.28
	2 (late April)	5	15.10	28.68	30.86	2.511	0.29
	3 (mid-May)	6	13.14	27.65	28.54	2.591	0.30
	4 (late May)	5	11.15	21.66	22.14	1.883	0.27
	*p* value	<0.001	<0.001	<0.001	<0.001	0.002	0.273
	l.s.d. (*p* < 0.05)	0.881	1.433	2.561	2.430	0.445	ns
WWAI2018	1 (mid-April)	7	22.68	39.22	41.33	3.905	0.36
	2 (late April)	7	22.58	38.22	40.11	3.610	0.42
	3 (mid-May)	6	21.65	31.71	35.08	3.411	0.46
	4 (late May)	6	19.45	28.95	32.77	2.606	0.50
	*p* value	0.51	<0.001	0.003	<0.001	<0.001	<0.001
	l.s.d. (*p* < 0.05)	ns	1.026	3.976	2.343	0.228	0.022
WWAI2019	1 (mid-April)	10	24.98	38.25	31.92	3.247	0.06
	2 (late April)	8	22.87	35.19	35.29	2.955	0.18
	3 (mid-May)	7	19.59	30.21	31.08	2.639	0.29
	4 (late May)	6	16.87	26.25	28.37	2.037	0.34
	*p* value	0.164	0.015	<0.001	0.052	<0.001	<0.001
	l.s.d. (*p* < 0.05)	ns	4.287	1.406	4.462	0.212	0.042
LFS2018	1 (mid-April)	12	18.41	48.58	40.62	7.562	0.21
	2 (late April)	10	18.33	39.52	37.29	6.789	0.31
	3 (mid-May)	9	15.96	45.91	33.50	6.346	0.38
	4 (late May)	8	15.17	41.62	31.92	5.424	0.44
	*p* value	0.02	0.201	0.335	0.002	<0.001	<0.001
	l.s.d. (*p* < 0.05)	2.119	ns	ns	3.064	0.534	0.029
LFS2019	1 (mid-April)	5	25.60	50.99	51.67	7.453	0.10
	2 (late April)	7	23.31	48.10	43.58	6.583	0.17
	3 (mid-May)	6	17.29	34.52	32.25	4.953	0.25
	4 (late May)	4	14.79	27.86	28.29	3.836	0.31
	*p* value	<0.001	<0.001	<0.001	<0.001	<0.001	<0.001
	l.s.d. (*p* < 0.05)	0.925	2.721	6.430	2.515	0.407	0.024
YAI2018	1 (mid-April)	11	13.27	30.61	32.64	4.051	0.18
	2 (late April)	11	12.26	28.72	28.42	3.617	0.30
	3 (mid-May)	8	13.46	26.62	26.94	3.484	0.39
	4 (late May)	7	13.77	24.35	25.89	3.086	0.39
	*p* value	<0.001	0.51	0.004	<0.001	0.3	<0.001
	l.s.d. (*p* < 0.05)	1.128	ns	2.995	2.618	ns	0.029

ns = not significant (l.s.d. not calculated).

**Table 4 plants-12-00474-t004:** Influence of sowing date on lentil grain yield and yield components. * = data not collected. TARC2018 = Trangie Agricultural Research Centre 2018 experiment; WWAI2018 = Wagga Wagga Agricultural Institute 2018 experiment; LFS2018 = Leeton Field Station 2018 experiment; YAI2018 = Yanco Agricultural Institute 2018 experiment; TARC2019 = Trangie Agricultural Research Centre 2019 experiment; WWAI2019 = Wagga Wagga Agricultural Institute 2019 experiment; and LFS2019 = Leeton Field Station 2019 experiment.

		Filled Pods	Unfilled Pods	Pod Number	Seeds Per Pod	Seeds Per Plant	100 Grain Weight (g)	Grain Yield (t/ha)	Machine Grain Yield (t/ha)
TARC2018	1 (mid-April)	9.7	1.8	11.5	1.2	11.4	4.35	1.609	*
	2 (late April)	6.3	1.2	7.5	1.2	7.3	4.19	1.541	*
	3 (mid-May)	6.8	1.2	8.0	1.1	7.3	4.15	0.986	*
	4 (late May)	4.5	0.6	5.1	1.1	4.5	4.13	0.678	*
	*p* value	<0.001	<0.001	<0.001	0.006	<0.001	0.023	<0.001	
	l.s.d. (*p* < 0.05)	1.8	3.1	1.9	0.077	18.896	0.141	0.26	
TARC2019	1 (mid-April)	39.3	23.5	62.6	0.7	47.2	3.93	0.789	*
	2 (late April)	23.3	6.8	30.0	1.0	28.9	3.95	0.730	*
	3 (mid-May)	28.7	8.6	37.3	0.9	34.9	3.89	0.831	*
	4 (late May)	19.7	9.3	29.0	0.7	22.2	3.84	0.544	*
	*p* value	<0.001	<0.001	<0.001	<0.001	<0.001	0.372	0.014	
	l.s.d. (*p* < 0.05)	6.244	3.345	8.441	0.088	8.799	ns	0.185	
WWAI2018	1 (mid-April)	25.3	9.8	35.1	1.4	35.4	4.59	1.380	1.200
	2 (late April)	29.0	6.1	35.0	1.4	39.9	4.58	1.506	1.310
	3 (mid-May)	25.3	3.7	28.9	1.3	32.2	4.52	1.553	1.270
	4 (late May)	27.5	4.8	32.3	1.2	33.3	4.60	1.310	0.910
	*p* value	0.747	0.011	0.209	<0.001	0.528	0.376	0.025	<0.001
	l.s.d. (*p* < 0.05)	ns	3.022	ns	0.068	ns	ns	0.149	0.072
WWAI2019	1 (mid-April)	11.7	1.9	13.6	0.8	12.0	2.58	0.187	0.220
	2 (late April)	15.8	2.2	18.0	1.1	18.8	3.11	0.519	0.541
	3 (mid-May)	15.5	3.1	18.5	1.1	19.2	3.41	0.763	0.560
	4 (late May)	18.9	1.8	20.7	1.0	19.9	3.52	0.696	0.384
	*p* value	0.134	0.269	0.085	0.053	0.119	0.008	<0.001	0.002
	l.s.d. (*p* < 0.05)	ns	ns	ns	0.172	ns	0.324	0.064	0.109
LFS2018	1 (mid-April)	33.8	14.8	48.6	1.2	41.1	4.71	1.587	1.363
	2 (late April)	29.3	10.2	39.5	1.3	37.9	4.70	2.080	1.853
	3 (mid-May)	34.5	11.4	45.9	1.4	48.5	4.45	2.433	2.158
	4 (late May)	34.4	7.3	41.6	1.4	48.6	4.31	2.356	2.033
	*p* value	0.424	0.004	0.335	<0.001	0.641	0.034	<0.001	<0.001
	l.s.d. (*p* < 0.05)	ns	3.204	ns	0.083	ns	0.254	0.207	0.202
LFS2019	1 (mid-April)	21.8	2.7	24.5	1.0	25.3	3.24	0.716	0.649
	2 (late April)	32.4	3.3	35.7	1.1	39.5	3.35	1.171	0.967
	3 (mid-May)	31.8	3.5	35.3	1.1	38.2	3.40	1.255	1.058
	4 (late May)	25.0	4.1	29.1	1.0	29.5	3.60	1.168	0.875
	*p* value	0.021	0.197	0.116	0.869	0.159	0.216	<0.001	0.006
	l.s.d. (*p* < 0.05)	7.015	ns	ns	ns	ns	ns	0.153	0.173
YAI2018	1 (mid-April)	21.2	11.2	32.1	0.6	12.2	3.86	0.761	0.492
	2 (late April)	25.9	9.2	35.1	0.6	15.2	4.06	1.119	0.631
	3 (mid-May)	29.7	6.6	36.3	0.6	18.8	4.11	1.399	0.734
	4 (late May)	27.2	5.8	33.1	0.7	17.7	3.91	1.168	0.683
	*p* value	0.086	0.026	0.384	0.013	0.027	0.032	<0.001	0.252
	l.s.d. (*p* < 0.05)	ns	3.621	ns	0.046	3.993	0.173	0.231	ns

ns = not significant (l.s.d. not calculated).

**Table 5 plants-12-00474-t005:** Lentil genotypes and their classification/characteristics evaluated in the 2018 and 2019 seasons.

Variety	Maturity Type	Seed Classification	Herbicide Tolerance (Imidazolinone)
PBA Ace	Mid	Medium red	No
PBA Blitz	Early	Medium red	No
PBA Bolt	Early/Mid	Medium red	No
PBA Hallmark XT	Mid/Late	Medium red	Yes
PBA Hurricane XT	Mid	Small red	Yes
Nipper	Mid	Small red	No
PBA Jumbo2	Mid	Large red	No
PBA Greenfield	Mid/Late	Large green	No

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
