# Peer review of "Effect of Sowing Date and Environment on Phenology, Growth and Yield of Lentil (Lens culinaris Medikus.) Genotypes"

_plants, 2023, doi:10.3390/plants12030474_

Round 1
Reviewer 1 Report
The paper investigates the phenological development of 8 lentil genotypes sown for 2 consecutive years at 4 sowing dates in 3-4 sites. Statistical analysis tests for main effects and interaction of the four factors.
The paper is almost clearly written, the topic is in line with the scope of the journal and has worldwide importance. The topic well introduced, and the methods clearly described.
However, I think that the lack of any biomass or yield data, make the great efforts in recording phenological development inconsistent, as the reader cannot understand if the differences in the length of phases influence plant growth and seed production, which are crucial drivers for the adoption of any novel cultural practice.
Authors measure and discuss the length of phenological phases in terms of GDD, without showing the equation they used and citing any literature (see McMaster and Wilhelm 1997). Moreover, they include a Tb of 0 °C for all phases, which probably is not true, at least for reproductive phases (see Porter and Gawith, 1999). I feel the importance of using GDD in plant phenotyping id not clear (see Russelle et al. 1984 and Arduini 2010 and 2016). GDD are generally used to eliminate site, or sowing date differences, so to reveal the temperature and daylength requirements of species and to emphasize genotypic differences. Thus, a genotype should need the same GDD at all sites and sowing dates, given all other limiting factors are removed.
Refer to these papers for more information:
McMaster, G.S., Wilhelm, W.W. 1997. Growing degree-days: one equation, two interpretations. Agric. For. Meteorol. 87:291–300.
Porter, J.R., Gawith, M. 1999. Temperatures and the growth and development of wheat: a review. Eur. J. Agron. 10:23–36.
Russelle et al., 1984. Crop Science, 24:28-32.
Arduini I., Ercoli L., Mariotti M., Masoni A. 2010. Coordination between plant and apex development in Hordeum vulgare spp. distichum. Comptes Rendus Biologies, 333:454-460 (IF 2010 1,603). doi:10.1016/j.crvi.2010.01.003.
Arduini I., Masoni A., Mariotti M. 2016. A growth scale for the phasic development of common buckwheat. Acta Agriculturae Scandinavica, Section B - Soil and Plant Science, 66(3):215-228. doi:10.1080/09064710.2015.1087587.
More comments in the text.
All summarized, in my opinion the paper cannot be published in this form and need major revisions, which consist in:
· Adding yield data, such as biomass, seed yield, harvest index.
· Changing the Tb for different phases and including upper threshold temperatures, so to uniform as possible the GDD needed for each genotype at all locations and eventually sowing date, if daylength or other factor do not also play a role.

Reviewer 2 Report
Introduction
Lines 72-80 – here I miss references.
Lines 82-90 – this needs a reference.
Methodology
Main comment – you are using 4 localities and ONLY 2 years of observations. I think this is a really short time for phenological observations. In the case of locality Yanco you have only 1 year of observations. I´m not sure how you can calculate the trends or correlations when you have only 2 years of phenological monitoring. I think you need to do more observations or use data from other years to analyze and calculate the best predictor for each phenological stage. I´m also wondering why are you analyzing years 2018 and 2019 in 2022. Did you observe the phenology also in 2020, 2021 and 2022? If you want to say something about the phenological response of plants you need at least 15-20 years of observation, but in other phenological studies, the shortest series are ca 25 years. Please, change the input phenological data, use phenological observations also from other years and then we can see the drivers for each phenological phases.
Table 4 – it will be helpful to indicate the years (for example in the first column).
Results
Line 129-139 – very confusing. It is very difficult to understand what the results showed. In this text, you are just repeating what is already in table 2 and you are speaking only about nonsignificant trends. Why you do not speak also about significance? At least I recommend describing the most important results from table 2 and changing the wording.
Table 2 – very confusing. I recommend preparing the table in a landscape format.
Table 2 – how do you calculate the trends or correlations?
Figure 2 – the legend and the description of the pictures is hardly readable and I cannot see the differences and meaning for a, b, c, d, e, f and g.
Figure 2 - In picture „a“ you are using different colors and these colors are not described in the legend. Figure 2 – I also do not understand the meaning of 4 different pictures in each part – different localities?
Figure 2 – In picture „c“ you left the description for y-axis.
Round 2
Reviewer 1 Report
Dear Authors,
the manuscript is acceptable for publication after your revision.